# Representation Learning of Structured Data for Medical Foundation Models

**Vijay Prakash Dwivedi**[1]    **Viktor Schlegel**[2,3*]   **Andy T. Liu**[1]

**Thanh-Tung Nguyen**[1*]    **Abhinav Ramesh Kashyap**[4*]

**Jeng Wei**[5]    **Wei-Hsian Yin**[5]    **Stefan Winkler**[1]    **Robby T. Tan**[1]

[1]ASUS Intelligent Cloud Services (AICS), Singapore
[2]Imperial College London, Imperial Global, Singapore
[3]University of Manchester, United Kingdom
[4]Crayon Software, India
[5]Cheng Hsin General Hospital, Taiwan

**Editors:** Marco Fumero, Clementine Domine, Zorah Lähner, Donato Crisostomi, Luca Moschella, Kimberly Stachenfeld

## Abstract

Large Language Models (LLMs) have demonstrated remarkable performance across various domains, including healthcare. However, their ability to effectively represent structured non-textual data, such as the alphanumeric medical codes used in records like ICD-10 or SNOMED-CT, is limited and has been particularly exposed in recent research. This paper examines the challenges LLMs face in processing medical codes due to the shortcomings of current tokenization methods. As a result, we introduce the `UniStruct` architecture to design a multimodal medical foundation model of unstructured text and structured data, which addresses these challenges by adapting subword tokenization techniques specifically for the structured medical codes. Our approach is validated through model pre-training on both an extensive internal medical database and a public repository of structured medical records. Trained on over 1 billion tokens on the internal medical database, the proposed model achieves up to a 23% improvement in evaluation metrics, with around 2% gain attributed to our proposed tokenization. Additionally, when evaluated on the EHRSHOT public benchmark with a 1/1000 fraction of the pre-training data, the `UniStruct` model improves performance on over 42% of the downstream tasks. Our approach not only enhances the representation and generalization capabilities of patient-centric models but also bridges a critical gap in representation learning models' ability to handle complex structured medical data, alongside unstructured text.

## 1 Introduction

In medical and healthcare domain, the extensive use of both unstructured text and structured alphanumeric data, such as medical codes, is prevalent for recording patient diagnoses, procedures, and

---

*Work performed while at ASUS Intelligent Cloud Services (AICS).

Proceedings of the II edition of the Workshop on Unifying Representations in Neural Models (UniReps 2024).

clinical details [13, 12, 27]. In particular, medical codes, including the widely used ICD-10 codes [26], are instrumental in the systematic recording of patient's health trajectory, often resulting in a longitudinal data of structured elements. Despite the advancements in large language models (LLMs) [1, 23, 7], the ability to effectively represent and understand these non-textual, structured data remains a challenge [21, 8]. This is particularly pronounced in multimodal data integration, where structured data like medical codes must be cohesively combined with unstructured clinical text notes to develop comprehensive and accurate patient models [25].

For the purpose of modeling non-textual data, one of the core limitations of current LLMs arise from their tokenization strategies [14], which are predominantly optimized for natural language processing (NLP), that include unstructured text. The tokenization methods, such as subword tokenization, fail to capture the hierarchical and compositional structure of medical codes, leading to inefficiencies in the model's ability to represent and interpret these codes accurately [4]. Recent studies [21, 8, 14] have shown that even the most advanced LLMs struggle with tasks involving the extraction and interpretation of medical codes, often performing worse than smaller models fine-tuned specifically for these tasks [18]. This limitation highlights a significant gap in the current capabilities of LLMs, particularly in handling the complex, structured nature of medical data. A sample of a medical record is presented in Section 4.1.

Another potential approach is to perform text-based LLM pre-training on alphanumeric medical codes. Subword tokenization, which has become a useful ingredient in several state-of-the-art LLMs [2], offers a mechanism for breaking down complex terms into smaller components. In the case of structured medical codes, such tokenization might involve splitting a hypothetical medical code like `AC310` into components such as `AC` and `310`, or `AC` and `3` and `10`, among other statistical combinations [14]. Such a strategy could be detrimental to the representation of medical codes, as these codes have distinct meanings when considered as a whole, despite the structural rules governing their individual characters and numbers.

In this work, we address the challenges of representing structured data, particularly medical codes, for foundation model pre-training by adapting and optimizing tokenization methods. Our approach involves adapting subword tokenization methods, such as byte pair encoding (BPE), to represent groups of medical codes that statistically co-occur as single tokens. This is similar to how subword units in a language that frequently appear together are treated as single tokens [2], as in Figure 1. We employ this tokenization method to create a custom tokenizer for our pre-training corpus, which encodes a patient's longitudinal data [27], consisting of structured medical codes assigned across multiple visits, as in Figure 2. In analogy to LLM or NLP concepts, we treat a patient as a

| Step 1: | Step 2: | Step 3: |
|---|---|---|
| Original: aaabdaaabac | Original: ZabdZabac | Original: ZYdZYac |
| Replace: aa with Z | Replace: ab with Y | Replace: ZY with X |
| Encoded: ZabdZabac | Encoded: ZYdZYac | Encoded: XdXac |
| Replacements: Z=aa | Replacements: Y=ab, Z=aa | Replacements: X=ZY, Y=ab, Z=aa |

Figure 1: An example illustration of Byte Pair Encoding (BPE) for an input word `aaabdaaabac`, with the steps that merges frequently co-occuring sub-words (`Z`, `Y`, `X`).

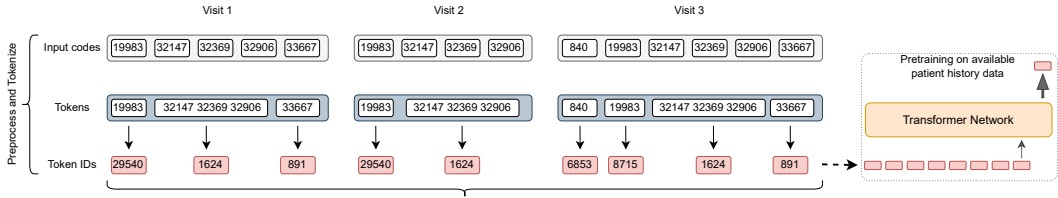

Figure 2: Illustration of the tokenization adopted for structured history data. All available structured data in a database is first passed to a Byte Pair Encoding (BPE) trainer to generate a custom tokenizer which is used to encode a patient's visit timeline, as illustrated with an example visit of a patient with 3 visits. The example shows a single token ID for multiple codes which occur frequent statistically, for instance the codes 32147  32369  32906 are encoded as a single token with the token ID 1624.

sentence, a visit as a word, and a medical code as a character, and process our custom tokenization step accordingly.

We conduct large-scale pre-training with the proposed `UniStruct` architecture (see Figure 3) for developing Medical Foundation Models, on two datasets—one internal and one public—and evaluate our approach across 16 downstream tasks. These datasets were sourced from both internal medical databases and publicly available structured medical records. On internal data sourced from Cheng Hsin General Hospital, our method significantly outperforms existing methods by upto 23% in recall scores, and on the public EHRSHOT dataset [27], despite a smaller pre-training corpus released publically[2], we achieve comparable or better results on over 42% of the tasks. We believe this advancement (i) improves the state-of-the-art methods to represent stuctured data, such as medical codes, and (ii) enables the adaptation of techniques which are critical to LLMs' success to patient-centric foundation models for healthcare applications.

## 2   Related Work

In this section, we review the closely related literature on structured data representation, the treatment of LLMs on medical codes and the recent foundation models on medical data and identify their limitations which we address through our work.

**Structured Data Representation.** The representation of structured data in healthcare domain, particularly medical codes such as ICD-10, remains a significant challenge due to the hierarchical and compositional nature of these codes, while being elementary in their definition [26, 4, 18]. Even the recent LLMs have struggled to effectively handle the unique requirements of such data [21, 8, 14]. [10] highlight similar issues in the domain of continuous numerical data, where standard tokenization methods like Byte Pair Encoding (BPE) [19] often fail as these would potentially split non-aligned tokens leading poor numerical operations. To address this, [10] propose a custom tokenization strategy that separates digits by spaces, treating each digit as an individual token, which significantly improves model performance in time series forecasting. This points towards a fundamental issue: both structured medical codes and continuous numerical data require specialized tokenization and representation designs to ensure the model's ability to accurately learn and predict complex patterns.

**LLMs on Medical Codes.** LLMs like GPTs [3] have shown impressive capabilities in natural language processing but have revealed significant limitations when applied to tasks on medical coding, particularly with structured data like ICDs codes. [8] demonstrate that these models often produce incorrect predictions due to their inability to fully comprehend the hierarchical and compositional aspects of medical codes. This challenge is analogous to the difficulties observed in processing continuous numerical data [10]. Similar limitations are identified in [21, 14] calling the need for a specialized representation of medical codes, which we primarily attend to in this work.

**Foundation Models on Structured Medical Data.** Foundation models have been increasingly applied to healthcare domain due to their potential to unify both structured and unstructured data. MedPaLM [20], for instance, is a large-scale pre-trained and fine-tuned LLM specifically trained on unstructured medical text. The EHRSHOT benchmark, introduced by [27], represents the largest publicly available database of longitudinal patient records, serving as a critical evaluation framework for these models. In this context, the CLMBR-T-base model [22] has shown promise in handling structured data, outperforming traditional baselines across multiple EHRSHOT downstream tasks. However, this model processes medical codes as individual elements with a feature processing stage, which limits its ability to effectively capture the co-occurrence and relational dynamics of multiple medical codes that together describe a patient's condition, analogous to how subword representation captures linguistic patterns in LLMs.

In this paper, we address the challenges of structured data representation, particularly in patient-centric longitudinal settings where LLMs have been found to underperform. We develop a foundation model that surpasses the capabilities of existing architectures, as in [22, 27], offering enhanced representation of patient structured data.

---

[2]The foundation model pre-trained on EHRSHOT benchmark in [27] has 2.57M patients in their pre-training corpus which is not released publicly. The training split that is publicly released is a smaller set of only 2.3K patients, *i.e.*, $1000\times$ smaller in size.

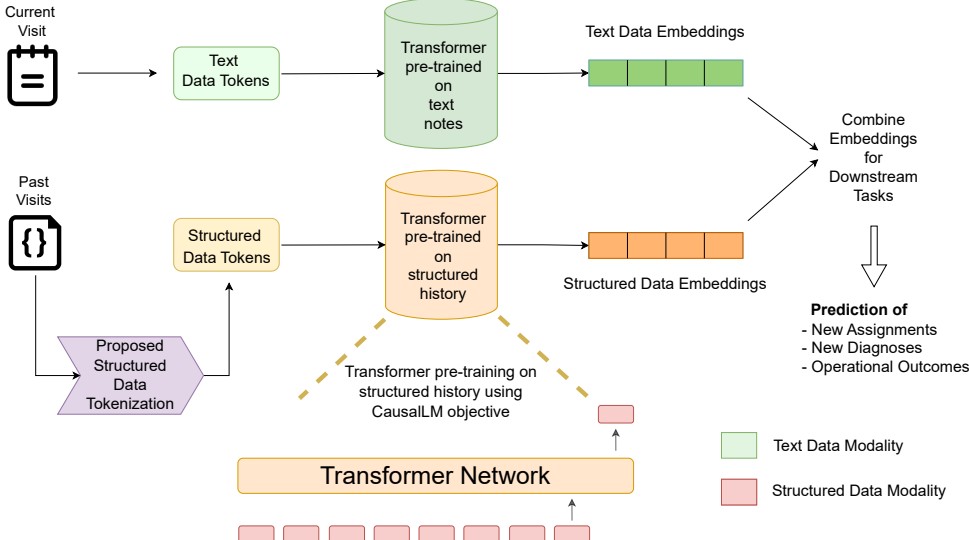

Figure 3: Overview of the `UniStruct` architecture. The architecture incorporates both text and structured data modality with the latter representing a patient's past history since structured data, such as medical codes, represent compressed, high-quality information about the patient's past visits. On datasets such as EHRSHOT [27], where only structured data is available, we use the structured data pipeline only. Note that the 'current visit' data consists of only 'clinical text' and not structured data, since part of structured data are the target of the model's predictions.

## 3    Proposed `UniStruct` Foundation Modal

In this section, we introduce our framework designed to address the challenges associated with representing structured medical data, specifically alphanumeric medical codes like ICD-10 codes [26], SNOMED codes [6] in large language models (LLMs). Our framework integrates structured data (e.g., medical codes) and unstructured data (e.g., clinical text) modalities, with a custom tokenization approach especially for structured medical codes. The tokenization process, as illustrated in Figure 2 is adapted from subword tokenization methods such as BPE [19, 2], allowing it to effectively represent the frequent co-occurrences and relationships inherent in patients' medical visits.

This inductive bias concerning the structured data enables the model to efficiently compress and consolidate the representation of a certain medical condition as the codes surrounding a condition has a high likelihood of being assigned together to a patient during a visit, as well as in future visits in conditions with long term occurence. We then pre-train a Transformer neural network [24, 16] on both data modalities constructed as respective sequences on large patient databases, both internal and public, and finetune on downstream evaluation tasks to evaluate the representations learnt.

### 3.1    Tokenization for Structured Data

A key innovation of our framework lies in the specialized tokenization process we developed for structured medical codes, leveraging the fact that the modeling of statistical co-occurences of textual units in LLMs are fundamental to their represenational power. Unlike the direct use of traditional tokenization methods optimized for natural language processing [14], our approach acknowledges the unique properties of medical codes, which represents a combination of alphanumeric elements and are as atomic as characters in texts.

The tokenization process begins by preprocessing the structured data, which includes ICD codes, exam codes, drug codes, SNOMED codes, and other medical identifiers from the patient's history. This data is analyzed to identify frequent co-occurrences and patterns that are critical for accurate representation. We employ Byte Pair Encoding (BPE) to segment the structured data into meaningful subwords or tokens. This method allows the model to capture both individual code elements and their frequent combinations, which are often indicative of common medical conditions. For instance,

the co-occurrence of specific drug codes with diagnosis codes can be effectively represented as a combined token, preserving the relational information between them. As an example illustration in Figure 2, the codes with code IDs 32147  32369  32906 occur together frequently and are tokenized together with the token ID 1624. Finally, the custom structured data tokenizer process transforms the structured data into a sequence of token IDs, which are then fed into a transformer network, as shown in Figure 3. For a direct comparison with CLMBR-T-base in [27], our approach differs in the aspect that CLMBR-T-base treats each element as a separate token is conceptually closer to character based tokenization, thereby losing the co-occurence patterns already available for use in the data. We show the comparison with such a tokenization in our experiments which we call 'Element Tokenization' where the element refers to the text-analogous 'Character'.

## 3.2   Model Architecture

Our framework employs a dual-modality transformer network, as shown in Figure 3, to integrate structured medical data with unstructured clinical text in one pipeline. In a data setting where no text records are available, such as on EHRSHOT[27], we only use the structured data Transformer.

**Text Modality Transformer.** For the text modality, we utilize standard language models, such as BERT-based models [5, 17][3]. This Transformer is pre-trained on clinical text notes and is responsible for processing unstructured text data, generating embeddings that capture the semantic information of the current visit's medical records. One might question why past clinical texts are not used. The reason is that structured medical codes offer better representations of past visits and compress the information that is comprised in a corresponding past visit.

**Structured Data Modality Transformer.** For the structured data modality, we use a separate Transformer which uses the custom tokenizer discussed earlier. It is pre-trained on structured history data using a Causal Language Modeling (CausalLM) objective [16], allowing it to learn the distribution and relationships of these tokenized representations. In other words, analogous to text pre-training, the longitudinal history of a patient can be understood as one sentence, where each *visit* is a *word*, and the *visit* consists of assigned *medical codes* which are the *characters* making up the *words*. As such, the sequential structure of a patient timeline is naturally encoded.

## 3.3   Downstream Prediction

The embeddings from both transformers are combined to create a comprehensive representation of the patient's medical state, including the current visit as well as past history. This combined representation is then utilized for various downstream tasks, such as predicting new assignments, diagnoses, or operational outcomes [27]. We adopt a logistic regression head for the downstream prediction.

# 4   Numerical Experiments

We now present our numerical experiments to evaluate the proposed `UniStruct` architecture on (i) a large scale internal dataset, and (ii) a public benchmark of longitudinal patient history, EHRSHOT [27]. We first describe the experimental setup including the details on the datasets, then show the results on various downstream tasks, including a New Medical Code Assignment task on the internal dataset, and 15 downstream tasks on the public EHRSHOT benchmark, and finally discuss our findings.

## 4.1   Datasets and Setup

**Internal Data.** We utilize an internal dataset from Cheng Hsin General Hospital[4], comprising of over 10 million records of over 765,000 patients multiple visits averaging 16.25 visits per patient. Consequently, each record of a patient, as shown in the de-identified sample below, includes a longitudinal timeline of visits, along with structured data such as ICD-10 codes, drug codes, and other medical identifiers for every visit. Additionally, we source the clinical text associated with these

---

[3]We do not use billion-parameter scale large language models (LLMs) for text modality here for efficiency purposes and to maintain the model parameters of both Transformers in Figure 3 at approximately 100 million.

[4]`https://chghims.com.tw`

records. However, for our experiments, we only use the clinical text from the current visit, as detailed in Section 3.2. The internal dataset includes approximately 1.2 billion pre-training tokens.

```
{
"EncounterDate": 20200808,
"CaseNo": "<redacted>",
"drugs": ["SAL03I", "SOL01O", "BR002O", "SOL02I", "KET05I",
         "DIM04O", "SOL01O", "KET05I", "VOL01O", ...],
"pid": "<redacted>",
"gender": "F",
"icds": ["R109", "M6080", ...],
"division": <redacted>,
"doctor": "<redacted>",
"soap": "low abdomial pain for 2 week and left flank
        pain today, dairrhea was also noted ...   ",
"birth_year": <redacted>,
"structured_history":
        {"20180111": ["J069", "J09X2", "00204", "01802", "05201",
                     "ALL02O", "MED02O", "PAN04O", ... ],
         "20180508": ["K2900", "R109", "00203", "01802", "05201",
                     "32006", "BIS01O", ...],
         "20190102": ["E860", "K529", "R109", "00204", "01802",
                     "05201", "06012", "06505", "08011", "08013",
                     "09005", "09015", "09017", "09021", "09022",
                     "09026", "32001", "32006", "39004", "B0029A",
                     "DIM04O", "GN0305B", ...]
        ...}
}
```

Given that our internal dataset contains both unstructured text and structured medical codes, we employ both modalities within the `UniStruct` architecture. The final embeddings of these two modalities are combined through a downstream head, which predicts New Medical Code Assignment, a multi-label classification task, for a patient during their current visit.

**Public Benchmark.** For public evaluation, we use the EHRSHOT dataset [27], a recent benchmark consisting of longitudinal EHR records that include diagnoses, procedure codes, prescription codes, and other medical identifiers. The publicly available dataset contains records for 6,739 patients, with approximately 2,300 patient records in the training split that we use for pre-training. Although this dataset is significantly smaller than our internal dataset, it is, to the best of our knowledge, the most suitable benchmark for publicly evaluating our proposed method. The dataset encompasses 15 downstream tasks, including 3 operational outcomes (such as predicting long length of stay, 30-day readmission, and ICU transfer); 5 lab test result predictions for conditions like thrombocytopenia and hyperkalemia; 6 assignments of new diagnoses ranging from acute MI to lupus; and 1 chest X-ray findings task covering 14 possible observations. Our primary focus is on the first 14 tasks, as the chest X-ray findings task includes 14 sub-tasks for which the available training corpus may not be sufficient to learn robust representations. Nonetheless, we still report our evaluations on the chest X-ray sub-tasks.

Since the EHRSHOT data consists solely of structured data, we utilize only the Structured Data Modality Transformer, as depicted in Figure 3, passing the code embeddings to a downstream head, following a similar implementation as in [27].

**Model Setup and Baselines.** In both evaluations, we maintain the model parameters at approximately 100 million, given the smaller size of our datasets compared to what is typically reported for state-of-the-art LLMs [7]. For the Transformer model used in the structured data modality, we follow the GPT-2 [16] configuration, similar to the approach adopted in [27]. We conduct pre-training and fine-tuning on a single A100 GPU server with 8 GPU cards.

For the baselines, we use a text only Transformer baseline for the internal evaluation, and follow the traditional baselines from [27] along with CLMBR-T-base [22] to compare our proposed `UniStruct` foundation model.

## 4.2 Results

We report the numerical results of our experiments on the internal dataset in Table 1 and Figure 4, while that on the EHRSHOT benchmark in Figures 5 and 6. Table 1 presents the comparison of different model settings on a New Medical Code Assignment task using the internal dataset, highlighting the performance metrics (Recall@5 and Recall@7) for models with and without structured patient

| Model Setting | Patient History | Recall@5 | Recall@7 |
|---|---|---|---|
| Text only | x | 0.497 | 0.561 |
| Text and Structured Data (Element Tokenization) | ✓ | 0.709 | 0.774 |
| Text and Structured Data (BPE Tokenization) - `UniStruct` **(Ours)** | ✓ | 0.725 | 0.792 |

Table 1: Comparison of different model settings and their performance on a new medical code assignment task on the internal dataset. The evaluation metric is Recall@K with higher being better. The 'Text only Transformer' has access to only the current visit's clinical text, while the 'Structured Data' includes the use of a patient's structured data from past visits. The 'Element Tokenization' denotes the treatment of each medical code separately, unlike BPE, as discussed in Section 3.1.

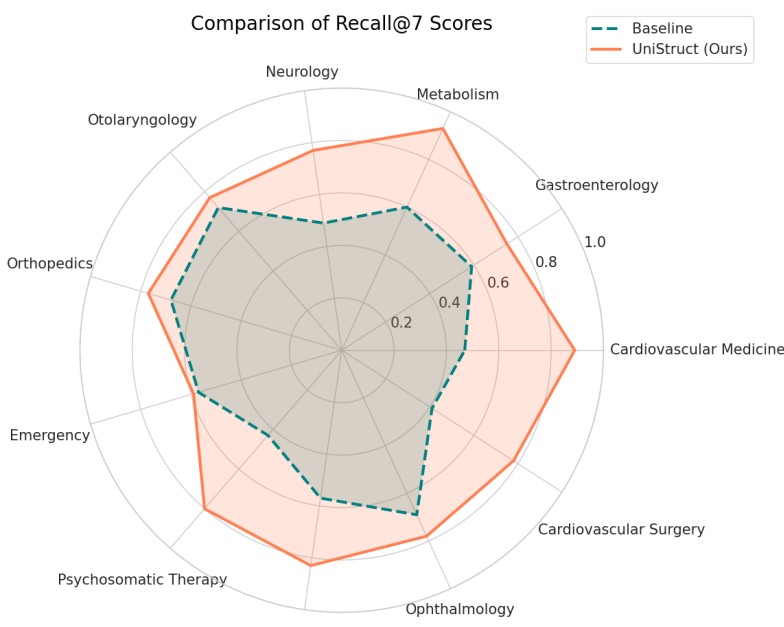

Figure 4: Department-wise evaluation on the new medical code assignment task on the internal dataset for the top 11 departments which cover more than two-third of all cases statistically. The Baseline model denotes the text only model without patient history as in Table 1.

history. Figure 5 showcases the `UniStruct` foundation model's performance across 14 downstream tasks from the EHRSHOT benchmark. Figure 6 details the model's results on various CheXpert subtasks [11] related to chest X-ray findings. Figure 4 offers a department-wise evaluation of the `UniStruct` model's performance on the new medical code assignment task, specifically across the top departments in the internal dataset which comprise over two-thirds of all clinical records in the database.

## 4.3 Discussion

The results from our internal evaluation highlight several key insights into the effectiveness of our proposed `UniStruct` model in handling structured medical data, particularly through the inclusion of patient history and the application of a custom tokenization strategy. On the public evaluation on EHRSHOT benchmark, despite the small pre-training corpus used by `UniStruct`, it outperforms or performs comparably on several downstream tasks.

### 4.3.1 Internal Evaluation

- **Impact of Patient History:** The inclusion of patient history in our model has shown a significant improvement over the baseline, which did not incorporate historical data. Specifically, we observed up to a 23% enhancement as reported in Table 1 in absolute

Recall@K scores for the predicting new medical code assignments. This finding is consistent with prior studies [27] that emphasize the importance of leveraging patient history for better generalization in downstream tasks.

- **Advantage of Proposed Tokenization:** Our comparison between element tokenization, as discussed in Section 3.1 which does not treat multiple medical codes during a condition as one single entity, and BPE tokenization reveals a notable performance gain. The use of a subword tokenization for structured data, which captures the relational dynamics of multiple medical codes representing the same patient condition, resulted in approximately a 2% improvement in performance as illustrated in the same Table 1. This suggests that BPE tokenization more effectively preserves the context and relationships inherent in medical codes compared to treating each code as an isolated element, as seen in models like CLMBR-T-base on the EHRSHOT benchmark.

- **Cases by Individual Departments:** A more detailed analysis of specific cases by individual medical departments, such as those involving long-term heart conditions in Cardiovascular departments, as shown in Figure 4, reveals that our proposed `UniStruct` model demonstrates a 42% improvement over the baseline, which did not account for patient history modeling. Significant improvements are also observed in departments like Metabolism and Gastroenterology, where patients often have long-term or high-risk factors, making historical context particularly valuable [9]. In contrast, the more moderate improvements in Neurology, Otolaryngology, and Orthopedics suggest that while patient history is somewhat beneficial, its impact may not be as pronounced due to the nature of the diagnoses in these departments. Overall, the results highlight the advantages of our approach for conditions with long-term characteristics. On the other hand, the Emergency Department, which handles patient cases related to emergency visits, shows negligible improvement when patient history is considered. This is understandable, as the triaging process in emergency situations is often preliminary, and a patient's longitudinal timeline may not significantly contribute to predicting future emergency medical code assignments.

### 4.3.2 Public Evaluation - EHRSHOT

Despite our foundation model being pre-trained on an approximately 1/1000 fraction of the patients compared to the CLMBR-T-base model in [27], *i.e.*, around 2.3K compared to 2.57M patients, our approach either improves or performs comparably (within 0.01 margin) on around 42% of the tasks (6/14) which is the primary group of downstream tasks we target.

- **Positive Results:** Figure 5 presents a performance comparison across various tasks. Among the three operational outcome tasks shown in Figure 5a—ICD Admission prediction, long length of stay (LOS) prediction, and 30-day readmission prediction—our proposed model significantly outperforms others on the first two tasks. This validates the efficiency of our `UniStruct`'s tokenization in representing structured data. In Figure 5c, `UniStruct` surpasses CLMBR-T-base in predicting the assignment of Acute MI (Myocardial Infarction) and Celiac, while performing comparably in predicting Hyperlipidemia and Pancreatic

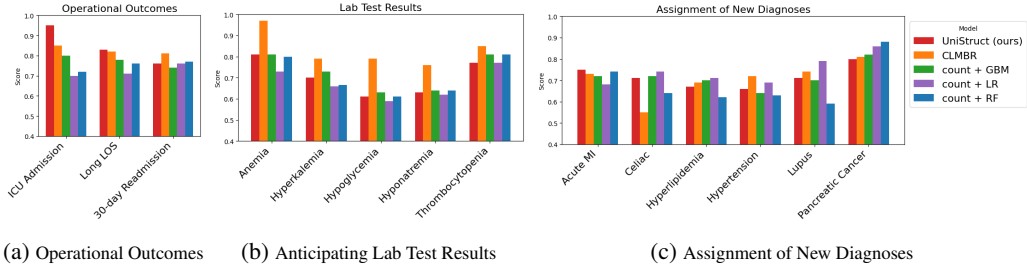

(a) Operational Outcomes     (b) Anticipating Lab Test Results     (c) Assignment of New Diagnoses

Figure 5: Comparison of different models on 14 EHRShot Downstream Tasks. The tasks are grouped into three categories: operational outcomes (left), anticipating lab test results (middle), and assignment of new diagnoses (right). The evaluation metric is AUROC, with higher scores indicating better performance. The proposed `UniStruct` model is denoted with 'red' color while the remaining models are baseline models following [27].

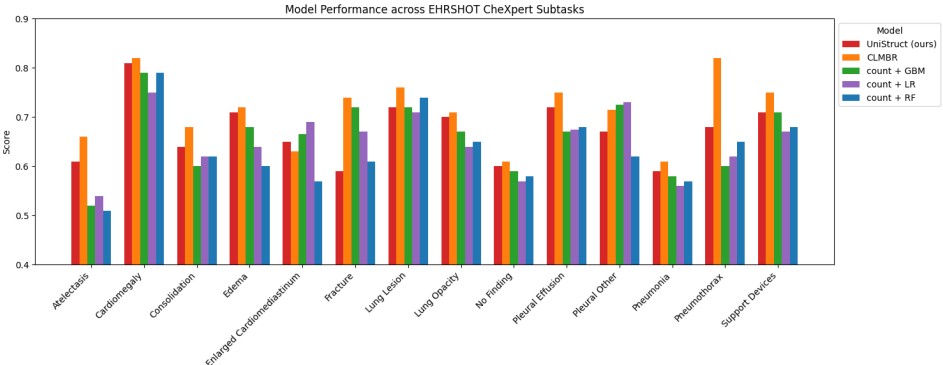

Figure 6: Comparison of different models on the EHRSHOT chest X-ray findings subtasks. Similar to Figure 5, the evaluation metric is AUROC, with higher being better. The proposed `UniStruct` model is denoted with 'red' color while the remaining models are the baselines following [27].

Cancer as new diagnoses. For the CheXpert findings task, we demonstrate comparable results on multiple subtasks, as illustrated in Figure 6. These include Cardiomegaly, Lung Opacity, and Pneumonia, among others. Notably, our model outperforms CLMBR-T-base on the Enlarged Cardiomediastinum subtask.

- **Negative Results:** As shown in Figure 5b, our model underperforms in relation to CLMBR-T-base by considerable margins on all lab test results tasks, despite showing comparable performance to the traditional feature based baselines. We hypothesize that lab test result prediction is relatively more complex, and since we only use 1/1000th of the CLMBR-T-base pretraining data, `UniStruct` likely misses important signals required to accurately model lab outcomes.

## 5 Conclusion

In this paper, we proposed a novel architecture designed to address the challenges associated with representing structured medical data, particularly medical codes, within language modeling-like framework. By integrating structured and unstructured data modalities and employing a custom tokenization approach adapted from subword tokenization methods like Byte Pair Encoding (BPE), we significantly improve the representation and generalization capabilities of patient-centric models. Our experiments, conducted on both an internal dataset and the public EHRSHOT benchmark, demonstrate that our method not only enhances model performance in tasks involving long-term patient conditions but also validates the effectiveness of our tokenization strategy in capturing the relational dynamics of medical codes. Despite limited pre-training data compared to what was used in existing models like CLMBR-T-base, our approach achieved comparable or better results across various downstream tasks on the EHRSHOT benchmark.

### 5.1 Limitations and Future Directions.

We believe the proposed `UniStruct` architecture, through its promising results on structured data modeling in this work, can advance multiple directions to develop more effective foundation models for multimodal settings with structured data as well as for use in the healthcare domain. First, there is the challenge of further unifying structured and unstructured data modalities, as the integration used in our work maintains two separate representation spaces for the two modalities considered. Next, the transfer of structured knowledge across different databases presents difficulties, particularly when dealing with varying internal structures and compliance requirements, which hinder the effective use of pre-trained tokens. Technically, the transfer of pre-trained knowledge from one database to another involves overcoming stark differences in the unique codes that may occur in individual databases. A recent work by [15] explores zero-shot transfer of tokenization, which can be relevant in such scenarios. Additionally, adapting the capabilities of state-of-the-art LLMs to our structured data representation is another challenge, as current approaches often require separate pre-training for each

modality, hindering unified learned representations. Addressing these limitations in future work will be essential for developing more robust and generalizable models in the healthcare domain.

## Acknowledgments and Disclosure of Funding

The authors would like to thank Kuluhan Binici, Jessen William and Cathy Chang for their constructive feedbacks and suggestions during the course of this work.

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
