# OpenReview forum: "Representation Learning of Structured Data for Medical Foundation Models"
_NeurIPS.cc/2024/Workshop/UniReps — UniReps_

### Official Review · Reviewer_WkkJ · 2024-10-05
**Evaluation of a Novel Tokenization Approach for Integrating Structured and Unstructured Medical Data**

**Rating:** 7
**Confidence:** 2

**Review:**

The paper introduces a novel method to integrate structured and unstructured medical data using a custom tokenization approach inspired by subword tokenization techniques. The proposed method shows promise, particularly in improving patient-related downstream tasks. One of the strengths of this work is its application to a private dataset, where it achieves notable performance gains.

However, the limited improvement observed on the public dataset raises questions about the overall contribution of the paper, as stronger results on diverse datasets would reinforce the method’s generalizability. Despite this, the paper is well-written, with clear explanations and justifications for each component of the model.

I do have a concern regarding the footer on the first page, which references NeurIPS conference. While this may be an oversight, it could potentially affect the submission’s reception, and I leave it to the AC to decide if this will impact the review process.
About the alignment with the scope of the workshop, it primarily focuses on representation learning, with an emphasis on integrating two data modalities (structured and unstructured).

---

> ### Author Response · Authors · 2024-11-06
> **Response to reviewer**
>
> We thank the reviewer for taking the time reading and evaluating our work, and are encouraged by the positive comments on the strengths of our approach. For the results on public datasets, we are constrained by the limited size of these datasets and report both the positive and negative outcomes. As suggested, we have updated the manuscript with the appropriate workshop footer note on the first page.

---

### Official Review · Reviewer_xqfN · 2024-10-06
**Representation Learning of Structured Data for Medical Foundation Models**

**Rating:** 6
**Confidence:** 4

**Review:**

The authors propose UniStruct Architecture that takes clinical text data and structured data to create a unified representation of data to be fed into foundation models.
From what I understand, the authors still use two different pre-trained transformer models to do the learning separately, hence how is this called a foundation model?
The authors could explain the model development and the architecture clearly. Moreover, there is no baseline comparison included to understand how the proposed model performs in comparison to other baselines on internal dataset.

---

> ### Author Response · Authors · 2024-11-06
> **Response to reviewer**
>
> We thank the reviewer for taking the time reading and evaluating our work.
>
> >**Reviewer**: From what I understand, the authors still use two different pre-trained transformer models to do the learning separately, hence how is this called a foundation model? The authors could explain the model development and the architecture clearly. Moreover, there is no baseline comparison included to understand how the proposed model performs in comparison to other baselines on internal dataset.
>
> **Authors**: We follow the usage of the terms as in Wornow et al., 2023. The model can be pre-trained in an unsupervised manner on large scale patient history datasets (as shown through both internal and public EHRSHOT datasets) and can be fine-tuned to perform on a wide range of clinical downstream tasks. As suggested for clarification on the architecture, we have included a new illustration (Figure 1) of the BPE process in the revised manuscript. In Table 1, for baseline comparisons on the internal dataset, we include (1) text-only Transformer models that do not incorporate structured data and (2) a second baseline that includes both text and structured data but does not use our custom tokenization. We then compare these with the proposed UniStruct model.
>
> [1] Wornow, M., Thapa, R., Steinberg, E., Fries, J. and Shah, N., 2023. Ehrshot: An ehr benchmark for few-shot evaluation of foundation models. Advances in Neural Information Processing Systems, 36, pp.67125-67137

---

### Official Review · Reviewer_k4W8 · 2024-10-07
**Review of "Representation Learning of Structured Data for Medical Foundation Models"**

**Rating:** 7
**Confidence:** 3

**Review:**

This paper presents a novel approach to address the challenges faced by large language models (LLMs) in representing structured medical data such as ICD-10 and SNOMED-CT codes. The authors propose a multimodal medical foundation model, UniStruct, which combines unstructured text and structured data. They introduce a custom tokenization technique specifically designed for structured medical data, significantly improving performance on internal and public benchmarks. The paper provides valuable insights into enhancing representation learning for structured medical data, but certain aspects of the methodology and evaluation require further clarification.

The quality of the work is high, providing a significant contribution to the field of medical AI by focusing on structured data representation, an area often overlooked in traditional LLMs. The use of custom tokenization tailored for medical codes and the multimodal approach to integrate structured and unstructured data is a step forward for medical foundation models.

Strengths:
The authors present a thorough and well-executed methodology, particularly in their custom tokenization approach for medical codes. The Byte Pair Encoding (BPE)-based technique captures the relational dynamics of codes effectively.
The experiments on both internal datasets and the EHRSHOT public benchmark validate the improvements, with significant gains in tasks such as medical code assignment.
Weaknesses:
The evaluation section could provide more in-depth comparisons with existing models, particularly on how UniStruct fares against other multimodal approaches in similar settings.
The paper lacks clarity on how pre-training on smaller datasets may generalize to larger-scale clinical environments. Further discussion of scalability would strengthen the findings.
Clarity
The paper is well-written overall, but a few sections could benefit from more clarity, especially for readers unfamiliar with the specifics of structured medical data and LLM integration.

Strengths:

The figures and illustrations, especially the tokenization process for structured data (Figure 1), are helpful in visualizing the model’s workflow.
The paper does a good job of explaining how structured medical data is integrated into the transformer model, providing clear distinctions between structured and unstructured modalities.
Weaknesses:

The introduction of custom tokenization for structured data is critical to the paper, but the explanation could be expanded to include more intuitive examples of how specific medical codes are tokenized and used.
The section on downstream prediction tasks is brief, and a clearer breakdown of how performance is measured on specific tasks would help the reader follow the results better.
Originality
The paper introduces original ideas, particularly the custom tokenization method for structured medical data, which addresses a significant gap in current LLM architectures. The approach of combining structured data with unstructured clinical text in one model pipeline is an important contribution to the development of patient-centric models.

Strengths:

The innovative use of BPE tokenization for structured medical codes stands out as a novel approach that could be applied beyond the medical field.
The architecture's ability to handle multimodal inputs (structured and unstructured data) provides a solid foundation for future work in the integration of different types of healthcare data.
Weaknesses:

Although the method is novel, it could be more explicitly compared to other multimodal LLM approaches that have been recently proposed. This would help to position the paper more clearly within the existing literature.
Significance
The significance of the work lies in its potential to greatly improve the use of structured medical data in large-scale machine learning models. The paper addresses a critical issue in medical AI—how to effectively represent and process structured data like medical codes, which are foundational in clinical workflows.

Strengths:

The results from the internal and public benchmarks (up to a 23% improvement in recall scores) demonstrate the real-world applicability of the UniStruct model in improving medical data representation.
The integration of structured and unstructured data has broad implications for enhancing patient care models, diagnostic tools, and other healthcare applications.
Weaknesses:

The model’s scalability and ability to generalize to larger, more diverse datasets remain unclear. Further testing on larger datasets beyond the provided internal and public benchmarks could highlight its broader applicability.
Pros
Innovative use of custom tokenization for structured medical data.
Significant improvements in medical code prediction tasks (up to 23% gain).
Strong methodology integrating both structured and unstructured data modalities.
Well-executed experiments on both internal and public benchmarks.
Cons
Limited comparison with other multimodal approaches.
Clarity could be improved, particularly in explaining tokenization and performance measurement.
Lack of discussion on the model’s scalability for larger datasets.
Minimal exploration of the model’s performance on more diverse tasks or clinical environments.

---

> ### Author Response · Authors · 2024-11-06
> **Response to reviewer**
>
> We thank the reviewer for taking the time reading and evaluating our work, and are encouraged by the positive comments of our approach and its applicability to the medical structured data. As suggested, we have incorporated a new illustration (Figure 1) of the BPE process in the revised manuscript, which can potentially benefit readers unfamiliar with the specifics of structured medical data and LLM integration.

---

### Official Review · Reviewer_zZxh · 2024-10-07
**Interesting work but results are not convincing**

**Rating:** 4
**Confidence:** 4

**Review:**

### Originality

- The paper proposes UniStruct, a transformer-based architecture that challanges structured data (medical codes) with unstructured data (clinical text). The proposed approach combines BPE tokenization of past visits with the embeddings of current clinical text.

### Quality

- The paper details the entire process, showcasing a thorough and well-documented methodology.

### Clarity

- Overall, this paper is clear and well-written, making it easy to follow. Each component of the methodology is sufficiently explained.

### Significance

- Pros:
    - The topic is relevant to the medical AI community.
    - UniStruct leverages both structured and unstructured data.
    - The proposed approach is validated in different settings.
- Cons:
    - The novelty is limited since the only innovation appears to rely on BPE tokenization. It is unclear if the encoding of previous visits is novel.
    - Compared to baselines (CLMBR-T-base), the proposed approach performs worse in most settings (Figures 4 and 5), even though the authors state in the abstract that the "UniStruct model improves performance on over 40% of the downstream tasks".
    - It is unclear why the authors trained on only a subset of the dataset used by CLMBR-T-base.
    Since there is still space in the paper, I suggest briefly explaining how BPE works.

---

> ### Author Response · Authors · 2024-11-06
> **Response to reviewer**
>
> We thank the reviewer for taking the time reading and evaluating our work. We have uploaded a revised version of the manuscript with the suggestions.
>
> >**Reviewer**: The novelty is limited since the only innovation appears to rely on BPE tokenization. It is unclear if the encoding of previous visits is novel. … I suggest briefly explaining how BPE works.
>
> **Authors**: In this work, we adapt subword-based BPE tokenization specifically to handle the structured data of alphanumeric medical codes. Prior research, as discussed in the paper, has shown that LLMs often struggle to effectively represent and interpret the structured, non-textual data prevalent in the medical domain. By using a custom tokenizer tailored for structured data, we address these limitations in our approach. As suggested, we have included a new illustration (Figure 1) of the BPE process in the revised manuscript.
>
>
> >**Reviewer**: Compared to baselines (CLMBR-T-base), the proposed approach performs worse in most settings (Figures 4 and 5), even though the authors state in the abstract that the "UniStruct model improves performance on over 40% of the downstream tasks".
>
> **Authors**: As reported in Section 4.3.2, our approach either improves or performs comparably (within 0.01 margin) on around 42% of the tasks (6 tasks out of 14).
>
>
> >**Reviewer**: It is unclear why the authors trained on only a subset of the dataset used by CLMBR-T-base.
>
> **Authors**: The CLMBR-T-base model is pretrained on a large internal dataset and, as noted in their paper, is not publicly available. The publicly accessible EHRSHOT dataset, which we use for pre-training, contains records for 6,739 patients, with approximately 2,300 patient records in the training split.

---

### Decision · Program_Chairs · 2024-10-10

**Decision:**

Accept

**Comment:**

In light of the positive reviewers' feedback and relevancy of the submission, we are pleased to accept this paper for presentation at UniReps 2024. We kindly ask the authors to incorporate the reviewers' suggestions and feedback in the final camera-ready version of the manuscript.